# Reorganisation of *Hoxd* regulatory landscapes during the evolution of a snake-like body plan

Isabel Guerreiro[1], Sandra Gitto[1], Ana Novoa[2], Julien Codourey[1], Thi Hanh Nguyen Huynh[1], Federico Gonzalez[1†], Michel C Milinkovitch[1], Moises Mallo[2], Denis Duboule[1,3]*

[1]Department of Genetics and Evolution, University of Geneva, Geneva, Switzerland; [2]Instituto Gulbenkian de Ciência, Lisbon, Portugal; [3]School of Life Sciences, Ecole Polytechnique Fédérale de Lausanne, Lausanne, Switzerland

**Abstract** Within land vertebrate species, snakes display extreme variations in their body plan, characterized by the absence of limbs and an elongated morphology. Such a particular interpretation of the basic vertebrate body architecture has often been associated with changes in the function or regulation of *Hox* genes. Here, we use an interspecies comparative approach to investigate different regulatory aspects at the snake *HoxD* locus. We report that, unlike in other vertebrates, snake mesoderm-specific enhancers are mostly located within the *HoxD* cluster itself rather than outside. In addition, despite both the absence of limbs and an altered *Hoxd* gene regulation in external genitalia, the limb-associated bimodal *HoxD* chromatin structure is maintained at the snake locus. Finally, we show that snake and mouse orthologous enhancer sequences can display distinct expression specificities. These results show that vertebrate morphological evolution likely involved extensive reorganisation at *Hox* loci, yet within a generally conserved regulatory framework.

*For correspondence: denis. duboule@epfl.ch

Present address: †Institute for Bioengineering of Catalonia, Barcelona, Spain

Competing interests: The authors declare that no competing interests exist.

## Introduction

Even though vertebrate species can display different morphologies, they all contain a strikingly similar repertoire of transcription factors and signalling molecules. In particular, genes with critical functions during embryonic development are often largely pleiotropic and highly conserved across species (for references, see *Kirschner et al., 2005*; *Duboule and Wilkins, 1998*). This universality of genetic and genomic principles has changed the evolutionary paradigm from the question of the nature of similarities to that of how distinct traits could evolve using such related developmental pathways (*Carroll, 2008*; *De Robertis, 2008*). Initially, *Hox* genes, as well as their structural and functional organization into genomic clusters were found well conserved across bilateria (*Duboule and Dolle, 1989*; *Akam, 1989*; *Garcia-Fernandez and Holland, 1994*; *Graham et al., 1989*; *McGinnis et al., 1984*). In addition, their mis-expression led to changes in the identity of both insect and vertebrate segments, illustrating their crucial role in the patterning of animal structures, even though the structures they specify are of very different nature in various taxa (e.g. (*Maeda, 2006*; *Lewis, 1978*; *Krumlauf, 1994*).

Tetrapods generally have four clusters of *Hox* genes (*HoxA, HoxB, HoxC* and *HoxD*), originating from genome duplications early in the vertebrate lineage (see e.g. *Lemons and McGinnis, 2006*) and located on different chromosomes, unlike fishes or some jawless vertebrates, which have more (*Prince et al., 1998*; *Mehta et al., 2013*; *Amores et al., 1998*). In addition, all vertebrate *Hox* clusters described thus far implement a particular type of regulatory process referred to as collinearity,

**eLife digest** Animals with a backbone can look remarkably different from one another, like fish and birds, for example. Nevertheless, these animals – which are also known as vertebrates – have many genes in common that shape their bodies during development. These genes include a family called the *Hox* genes, which control how an animal's body parts develop from its head to its tail and are needed to shape the animal's limbs. *Hox* genes are found clustered in groups within a vertebrate's DNA, and large regions of DNA on either side of a *Hox* cluster can, in some cases, physically interact with the *Hox* genes to regulate their expression.

So how do the same genes produce different body shapes? Different vertebrates regulate where and when their *Hox* genes are switched off and on in different ways. As such, it is likely that differences in gene regulation, rather than in the genes themselves, lead embryos to develop into the distinct shapes seen across the animal kingdom.

Snakes – for example – evolved from a lizard-like ancestor into elongated limbless animals as they have adapted to a burrowing lifestyle. However, it was not known if changes in how *Hox* genes are regulated have played a role in shaping the distinct body plan of snakes.

Guerreiro et al. have now compared how *Hox* genes are regulated in snakes, mice and other vertebrates, focusing on corn snakes and one particular cluster of *Hox* genes called the *HoxD* cluster. The comparison revealed that these *Hox* genes are regulated differently in developing snakes than in other vertebrate embryos. This is particularly the case for tissues that show the most differences when compared with other animals (such as the torso and genitals) or that are absent (such as the limbs). Although *Hoxd* genes are activated at different times and places in snakes than in other vertebrates, snake *Hox* genes appear to be regulated using the same general mechanisms as mouse *Hox* genes.

Guerreiro et al. suggest that changes to *Hoxd* gene regulation have contributed to the evolution of the snake's shape and have most likely influenced the body shapes of other vertebrates as well. However, the findings also suggest that these gene regulatory changes have been constrained by an existing regulatory mechanism that has been maintained throughout evolution. It remains for future work to address whether these changes in *Hox* gene regulation are a cause or a consequence of the snake's extreme body shape, or indeed a combination of the two.

whereby *Hox* genes are expressed sequentially in both time and space following their topological organization within each genomic cluster (*Gaunt et al., 1988*; *Izpisua-Belmonte et al., 1991*). This regulatory property is first observed during axial extension and, subsequently, in some structures such as the limbs (see (*Deschamps, 2007*; *Deschamps and van Nes, 2005*). In this latter case, and while the detailed underlying mechanism may be distinct from that at work in the major body axis (*Kmita and Duboule, 2003*), the general principle remains the same and was likely co-opted in the course of tetrapod evolution (*Spitz et al., 2001*), through the emergence of global enhancers located at remote positions on both sides of the cluster (*Lonfat et al., 2014*).

These complex regulations were extensively studied in the mouse, in particular at the *HoxD* locus, by using various targeted approaches in vivo. The *HoxD* cluster is surrounded by two gene deserts of approximately 1 Mb (megabase) in size, each one containing distinct sets of enhancers capable of activating specific sub-groups of target *Hoxd* genes depending on their location within the cluster. Each of these two gene deserts can be superimposed to a Topologically Associating Domain (TAD), i.e. a chromatin domain where DNA-DNA interactions in *cis* are privileged, for example between promoters and enhancers, and determined through chromosome conformation capture technologies (*Dixon et al., 2012*; *Nora et al., 2012*). The centromeric gene desert can activate the transcription of the *Hoxd9* to *Hoxd13* genes, whereas the telomeric gene desert, which is further subdivided into two sub-TADs (*Andrey et al., 2013*) controls the expression of *Hoxd1* to *Hoxd11* (see [*Lonfat and Duboule, 2015*]).

This bimodal regulation allows for the selected expression of *Hoxd* gene sub-groups in a series of secondary embryonic structures. During limb development, for instance, the telomeric TAD initially controls all genes from *Hoxd3* to *Hoxd11* in the proximal part of the limb bud, whereas more

posterior genes such as *Hoxd13* or *Hoxd12* are controlled subsequently in the most distal aspect of the incipient limb by enhancers located within the centromeric TAD (*Andrey et al., 2013*). This latter regulatory landscape also controls transcription of the same posterior genes during the outgrowth of external genitalia (*Lonfat et al., 2014*). Since snakes are limbless animals and they display highly specialized and divergent external genitals (*Tschopp et al., 2014*), the existence of such a bimodal type of regulation at the snake *HoxD* locus was uncertain. Therefore, we set out to investigate *Hox* gene regulation in snakes. While these animals cannot yet be considered as genuine model systems (*Guerreiro and Duboule, 2014*; *Milinkovitch and Tzika, 2007*), recent advances in their genomic analyses make their study increasingly interesting in an Evo-Devo context (*Castoe et al., 2013*; *Gilbert et al., 2014*; *Ullate-Agote et al., 2014*; *Vonk et al., 2013*). These analyses revealed that snakes, regardless of their extreme morphologies, have a tetrapod-like complement of *Hox* genes with only a few exceptions (*Vonk et al., 2013*; *Di-Poï et al., 2010*). Consequently, the serpentiform body plan may have evolved either along with changes in time and space of *Hox* gene expression or with a different interpretation of Hox protein functions (see [*Di-Poï et al., 2010*; *Woltering et al., 2009*]).

The analysis of *Hox* gene expression in the developing corn snake (*Pantherophis guttatus*) revealed a surprisingly well conserved collinear mRNA distribution along the anterior-posterior axis. However, the rather strict correlation between morphological landmarks and the anterior borders of *Hox* transcript domains, usually seen in mammals and birds, was not always present in snakes (*Burke et al., 1995*; *Woltering et al., 2009*) (see also *Head and Polly [2015]*). It was thus concluded that some Hox proteins had likely changed (part of) their functionality. In addition, the fact that the most posterior *Hox* genes were poorly expressed in the extending tailbud was tentatively associated to the unusually large number of segments (*Di-Poï et al., 2010*), together with an increased pace in segmentation (*Gomez et al., 2008*).

In this work, we used a combination of experimental approaches to try and elucidate the nature of the differences in *Hoxd* gene regulation between snakes and mice at comparable stages of their early development. We find that, even though the structural organization of the corn snake *HoxD* cluster resembles that of tetrapods, the extreme body plan observed in snakes is associated with an extensive regulatory restructuring. In snakes, mesodermal enhancers are mostly located inside the cluster itself, whereas other vertebrates make use of long-range regulations located at remote positions. In addition, we show that despite the loss of limbs, the bimodal chromatin organisation at the *Hoxd* locus found in tetrapods is conserved in the snake lineage. However, we find that the regulation of snake *Hoxd* genes during the development of the external genitalia is different from that of other tetrapods, even though the general logic is conserved. In this latter case, the change in enhancer activity from a limb to an external genital specificity seems to have occurred. Altogether, we conclude that *Hoxd* gene regulation in the snake is in many ways distinct from the situation in mammals. We discuss the possible causative nature of these changes in the evolutionary transformation towards a serpentiform body plan.

## Results

To analyse the regulation of the snake *HoxD* cluster, we initially had to complement the available genomic information (*Ullate-Agote et al., 2014*) with high coverage sequencing of the gene cluster itself, along with the two flanking gene deserts. We screened a corn snake custom-made BAC library using as probes DNA sequences conserved from mammals to the anole lizard, present within this large DNA interval. A scaffold was built out of 13 overlapping BACs, which were selected for sequencing and from which a 1.3 Mb large DNA sequence of high quality was obtained (*Figure 1— figure supplement 1A*). The structural analysis of the corn snake *HoxD* cluster revealed that, as for other species of snakes whose genomes were recently released (*Vonk et al., 2013*; *Castoe et al., 2013*), all *Hoxd* genes but *Hoxd12* are present and share the same transcriptional orientation within the cluster. When compared to the mouse, the corn snake cluster is about 1.5 fold larger (*Figure 1A*) (*Vonk et al., 2013*), likely due to a higher repeat content (*Di-Poï et al., 2009*) and consistent with the structures of the *HoxD* clusters of both the king cobra and the python (*Castoe et al., 2013*; *Vonk et al., 2013*).

We plotted the sizes of the mammalian, reptile, bird and fish *HoxD* clusters against the size of their respective genome. When reptiles were excluded from the linear regression analysis, an $R^2$

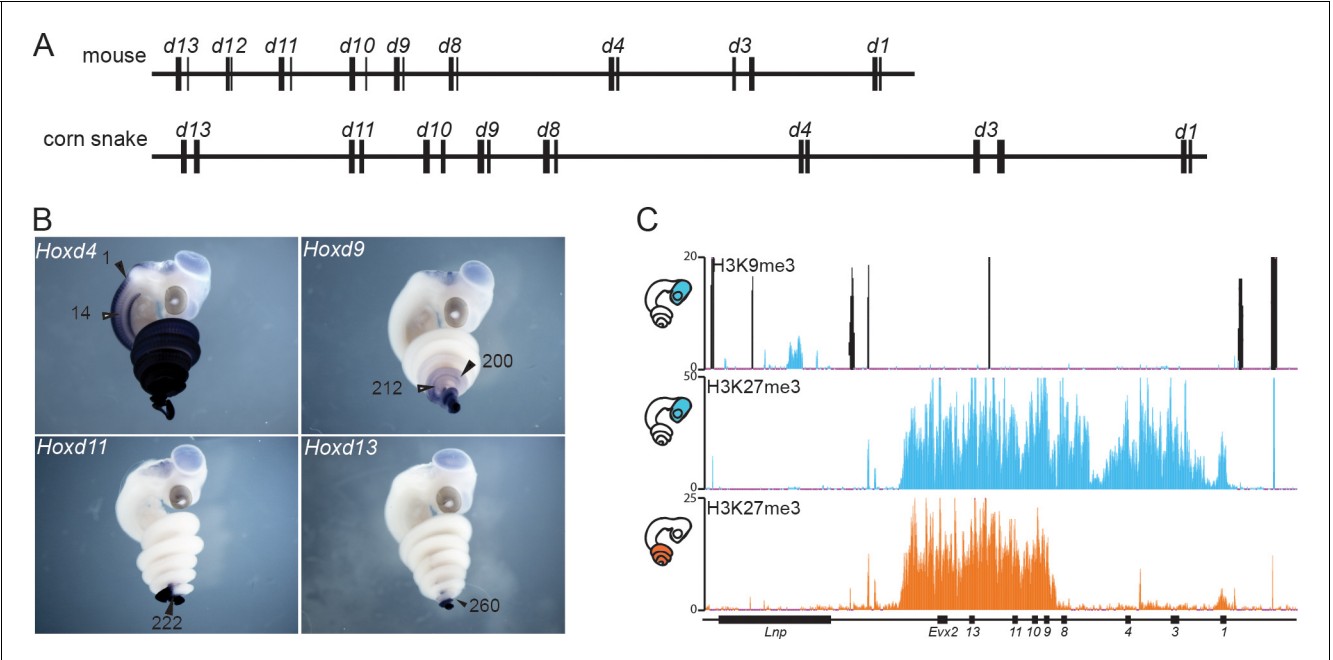

**Figure 1.** The snake *HoxD* cluster. (**A**) Schematic representation at the same scale of the mouse (top) and corn snake (bottom) *HoxD* clusters. Exons are represented by black rectangles. (**B**) Whole-mount in situ hybridization of corn snake embryos at 8.5 dpo (days post oviposition) showing expression of *Hoxd4, Hoxd9, Hoxd11* and *Hoxd13*. Numbers define the somite number where the most anterior levels of expression are detected. The black arrowhead points to the neural tube whereas the white arrowhead shows mesoderm. A single black arrowhead indicates that the neural and mesodermal boundaries coincided. (**C**) Detection of both H3K9me3 and H3K27me3 histone modifications by ChIP-seq in corn snake brain (top and middle tracks) and of H3K27me3 marks in the posterior trunk of 0.5–2.5 dpo snake embryos (bottom track). Blue is for brain and orange for posterior trunk, as schematized on the left. The black peaks in the top track represent artifactual signals also present in the input chromatin mapping.

The following figure supplements are available for figure 1:

**Figure supplement 1.** Corn snake *HoxD* cluster and surrounding regulatory landscapes.

**Figure supplement 2.** Sequence conservation at the *HoxD* locus.

value of 0.43 was scored, indicating significant correlation. However, when the corn snake, king cobra and Burmese python cluster sizes were added to the analysis, the $R^2$ value was reduced to 0.027 (*Figure 1—figure supplement 1B*). Even though snake Hox clusters show a size larger than what would be expected based on their genome size, the green anole lizard cluster is, from the vertebrate species analysed, the one with the lowest level of correlation with genome size ($R^2$=0.0012). When we performed the same correlation analysis for the regulatory gene deserts that surround the *HoxD* cluster, high values of $R^2$ were scored both by excluding and including squamate values. However, the size of the squamate 3' gene desert clearly showed a better correlation with genome size than the 5' gene desert (*Figure 1—figure supplement 1B*).

Because the increased size of the cluster in Squamata correlated with the presence of a high number of transposable elements (*Di-Poï et al., 2009, 2010*), we investigated the number and type of repeats present in the *HoxD* locus. We found that the corn snake cluster contains more than twice as many repeats as the mouse counterpart. In addition, while the mouse *HoxD* cluster is mainly composed of SINEs (short interspersed elements), the corn snake cluster is composed of different types of transposable elements including LINEs (long interspersed elements) and DNA transposons (*Figure 1—figure supplement 1C*). The 5' and 3' gene deserts that surround the cluster contain a similar amount of repeats in the two species. Both gene deserts in mice include a wider range of repeat element types than in the cluster itself, yet SINEs remain the most represented transposable elements, whereas snake gene deserts mostly included LINEs (*Figure 1—figure supplement 1C*). Our deep DNA sequence of the entire corn snake *HoxD* locus, including both flanking gene deserts, allowed a

global conservation analysis to be performed between non-coding regions amongst different vertebrate species. Surprisingly, we found that the pattern of conservation of the snake *HoxD* genomic landscape is almost identical to that of the chicken, when compared to the mouse sequence (*Figure 1—figure supplement 2*).

## The snake *HoxD* cluster

Because the silencing of transposable elements is often paralleled by the modification of histone H3 at lysine 9 (H3K9me3 (*Kidwell and Lisch, 1997*; *Martens et al., 2005*; *Friedli and Trono, 2015*), heterochromatin-like islands within the snake *Hox* clusters may be associated with severe modifications in gene regulation (*Di-Poï et al., 2009*; *Woltering et al., 2009*). H3K9me3 modifications are normally not found at *Hox* loci in tetrapods, which like many other genomic loci containing genes of importance for development, are also poor in transposons (*Simons et al., 2007*). Therefore, we performed a ChIP-seq experiment with an antibody against this histone modification on micro-dissected embryonic snake brain (*Figure 1*), a tissue that we routinely use as a negative control for *Hox* gene expression (*Figure 1B*). No particular H3K9me3 enrichment was scored over the length of the *HoxD*

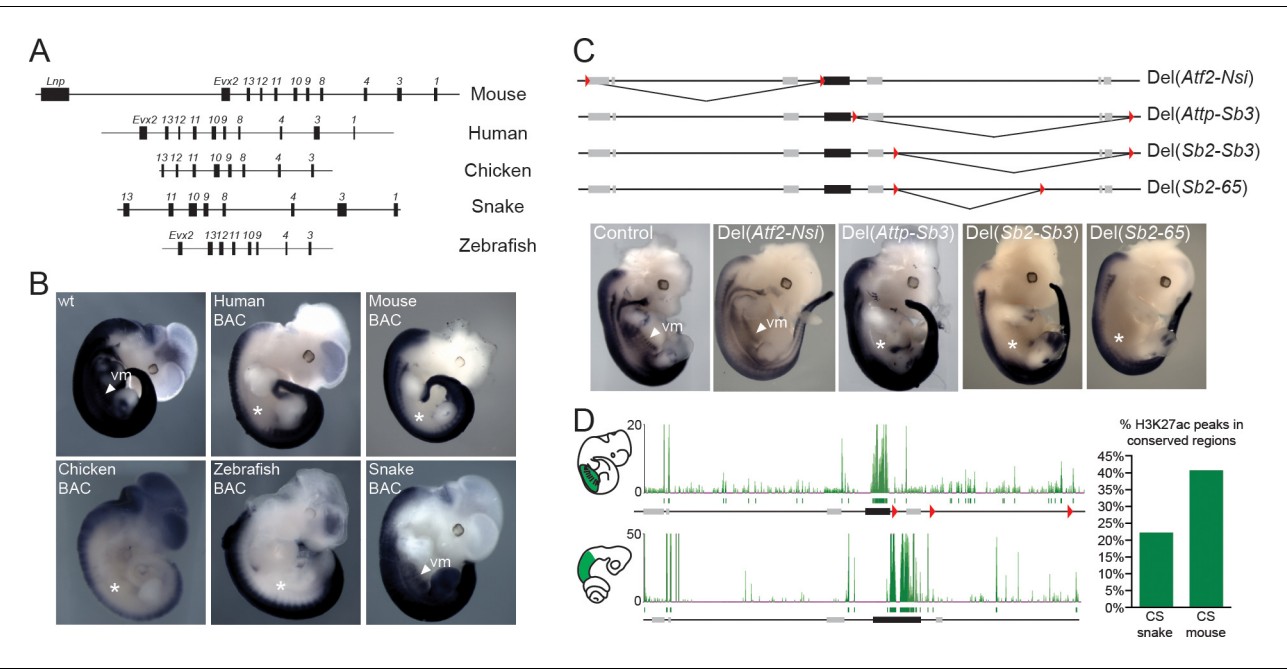

**Figure 2.** Location of *Hoxd* trunk mesodermal enhancers. (**A**) Schematic representation, at the same scale, of the mouse, human, chicken, corn snake and zebrafish BAC clones used to generate the transgenic mouse lines. Exons are represented by black rectangles. (**B**) Lateral view of whole-mount in situ hybridizations of *Hoxd4* using E11.5 mouse embryos transgenic either for the mouse, the human, the chicken, the zebrafish or the corn snake BAC. (**C**) Schemes illustrating the various deletion stocks (top) and whole-mount in situ hybridization of E12.5 mouse embryos with the *Hoxd4* probe in corresponding deleted mutant embryos (bottom). LoxP sites are indicated as red triangles, the *HoxD* cluster is represented by a black rectangle and other genes are shown with grey rectangles. vm indicates expression in the ventral mesoderm and white asterisks represent the absence of expression in this tissue. (**D**) ChIP-seq analysis over the mouse and snake *HoxD* loci of H3K27acetylation using anterior trunk mesodermal tissue of E11.5 mouse embryos and 5.5 dpo corn snake embryos (left). Green boxes under each ChIP-seq mapping represent peaks called by the MACS algorithm (*Zhang et al., 2008*). On the right, a graphical representation is shown of the percentage of conserved regions between the mouse and corn snake *HoxD* loci, which are enriched for H3K27ac in each species.

The following figure supplements are available for figure 2:

**Figure supplement 1.** Detailed analysis of the mesodermal enhancer activity in the 3' gene desert.

**Figure supplement 2.** Regulatory potential of the mouse 3'-located, telomeric gene desert in trunk mesoderm.

**Figure supplement 3.** Regulatory potential of a mesodermal enhancer sequence (MSS) located in the telomeric gene desert.

cluster and the closest located peak was identified in an intron of the *Lunapark* gene, i.e. at a position unlikely to have any critical impact on *Hoxd* gene expression (*Figure 1C*).

In tetrapods, the proper collinear regulation of *Hox* gene transcription was associated with the progressive removal of H3K27me3 coverage (*Soshnikova and Duboule, 2009*), a histone modification deposited by the Polycomb complex PRC2 (*Margueron and Reinberg, 2011*). We checked if this repressive system would operate similarly during the elongation of the snake body axis by performing an H3K27me3 ChIP-seq, either in the embryonic brain, or in a part of the posterior trunk excluding the post-cloacal region (*Figure 1C*). In the absence of *Hox* gene transcription (brain), the entire cluster was decorated with H3K27me3 marks, forming a dense domain of Polycomb repression as seen previously in other species. In contrast, the posterior trunk tissue displayed an H3K27me3 coverage specifically over the 5' part of the gene cluster, containing the most 'posterior' *Hoxd* genes (*Figure 1C*). In parallel, whole-mount in situ hybridization (WISH) to assess *Hoxd* gene expression revealed a clear correlation between the domain of active *Hoxd* genes and the absence of the H3K27me3 mark (*Figure 1B*). From these experiments, we concluded that both spatial collinearity and the associated dynamics of chromatin structure accompanying progressive gene activation are implemented in snakes as in any other vertebrate species studied thus far.

## Regulatory potential of the *HoxD* cluster in vertebrates

In tetrapods, regulatory elements controlling *Hox* gene expression are found at various positions. The mouse *HoxD* cluster for instance contains regulatory elements, which are mainly involved in driving *Hox* gene expression along the anterior-posterior body axis during gastrulation, whereas remote enhancers located outside of the cluster itself regulate transcription in other organs or structures (*Spitz et al., 2001*; *Lonfat and Duboule, 2015*). Therefore, to try and identify snake-specific differences in the modes of regulations, we compared the regulatory potential of the snake *HoxD* cluster with that of other vertebrates by using a BAC transgenic approach in mice, whereby BACs containing *HoxD* clusters of either human, mouse, chicken, snake and zebrafish were randomly integrated in the mouse genome (*Figure 2A*). The expression of *Hoxd4* was monitored by in situ hybridization with species-specific probes and, under these experimental conditions, all mammalian transgenic BACs showed transcript patterns restricted to the dorsal part of the main embryonic body axis (*Figure 2B*), resembling the pattern obtained when a single copy *Hoxd4/LacZ* transgene was used (*Tschopp et al., 2012*). Interestingly, however, this pattern represented only a subset of the full *Hoxd4* expression pattern as seen either by WISH on control embryos (*Figure 2B*), or on previous reporter *Hoxd4/lacZ* transgenes likely integrated as tandem repeats (*Zhang et al., 1997*). Indeed, expression was scored mostly in the neural tube, yet not in the ventral mesodermal tissues of the upper trunk, i.e. above the level of hindlimbs. To better determine which mesodermal components had their *Hox* gene expression affected in the isolated human BAC line, we performed a *Hoxd4* WISH in a sectioned embryo. We found that, at least at this stage of development, the human *Hoxd4* gene was expressed only in the neural tube (*Figure 2—figure supplement 1A*).

We then investigated the expression of *Hoxd4* from either the chicken or the zebrafish BAC transgenic lines and found a similar expression pattern, again mostly limited to the neural tube as well as the dorsal-most part of the somites (*Figure 2B*). Altogether, these results suggested that enhancers controlling the robust expression of *Hoxd* genes in various mesodermal derivatives are, for the most part, located outside of the cluster itself. Alternatively, some mesodermal enhancers could be located inside the *HoxD* cluster, yet they may require additional sequences located at remote positions to properly impact upon the transcription of target genes in physiological conditions. Consequently, we searched for the location(s) of such enhancers outside the *HoxD* cluster by using a set of targeted deletions flanking the locus on either side of it.

## Reshuffling mesodermal enhancers

We first analysed the expression of *Hoxd4* in mouse embryos lacking the centromeric TAD, which contains strong enhancers with digit and genital specificities (*Lonfat et al., 2014*). Mutant embryos carrying this *HoxD*^Del(Atf2-Nsi)^ deletion (*Montavon et al., 2011*) showed a domain of *Hoxd4* expression comparable to control embryos (*Figure 2C*; Del(*Atf2-Nsi*). In contrast, embryos carrying the *HoxD*^Del(Attp-Sb3)^ deletion of the opposite TAD, located telomeric to the *HoxD* cluster, which contains various enhancer sequences (*Andrey et al., 2013*; *Delpretti et al., 2012*), displayed reduced

amounts of mRNA steady state levels in the ventral mesoderm (*Figure 2C*; Del(*Attp-Sb3*). Consistently, the repositioning of potential telomeric enhancers several megabases far from the target genes, through the *HoxD^Inv(Attp-CD44)* inversion, displayed no clear expression in the ventral mesoderm of the thoraco-lumbar region (*Figure 2—figure supplement 2*). These results indicate that the telomeric gene desert contains most of the enhancers necessary for *Hox* gene expression in ventral mesoderm. However, unlike what was observed in the Human BAC line, *Hoxd4* expression in the *HoxD^Del(Attp-Sb3)* and *HoxD^Inv(Attp-CD44)* mutant lines was also scored in the dorsal-most part of the somites and not exclusively in the neural tube (*Figure 2—figure supplement 1A* and *2*).

To more precisely localize potential mesodermal enhancers within the deleted DNA interval, we used four additional mutant stocks carrying smaller deletions. Both the *HoxD^Del(Sb2-Sb3)* and the *HoxD^Del(Sb2-65)* mutant alleles (*Andrey et al., 2013*) resulted in expression patterns for *Hoxd4* similar to that obtained with the Del(*Attp-Sb3*) deletion of the entire gene desert (*Figure 2C*; Del(*Sb2-Sb3*), Del(*Sb2-65*), i.e. lacking any detectable expression in ventral mesoderm. In contrast, such mesodermal expression was scored in the smaller *HoxD^Del(65-Sb3)* and *HoxD^Del(Attp-Sb2)* deletion alleles (*Figure 2—figure supplement 2*; Del(*65-Sb3*), Del(*Attp-Sb2*). This set of analyses indicated the presence of mesodermal enhancer(s) within a segment of the telomeric gene desert. In addition, the distribution of H3K27ac modifications in the mouse trunk mesodermal tissue, a histone mark associated with putative active enhancers and promoters, was clearly enriched in the telomeric gene desert when compared to the centromeric counterpart (*Figure 2D*, top) with 18 significant peaks telomeric to the cluster versus only 7 located in the centromeric gene desert. We thus concluded that most trunk mesodermal enhancers acting over *Hoxd4* and presumably affecting other *Hoxd* genes, are located in the telomeric gene desert.

Because the regulatory sequences located in the telomeric TAD were described to globally drive concomitant expression of several genes located in the central part of the gene cluster rather than individual *Hoxd* genes (*Delpretti et al., 2013*; *Andrey et al., 2013*), we also analysed the expression of both *Hoxd3* and *Hoxd8* in the absence of the telomeric gene desert. Similar to *Hoxd4,* the expression of these two other *Hoxd* genes was lost in the ventral mesoderm (*Figure 2—figure supplement 1B*) thus suggesting that the telomeric gene desert contains sequences necessary for the expression of multiple *Hoxd* genes in the ventral mesoderm of the upper trunk.

## Reorganization of mesodermal enhancers in the snake *HoxD* locus

Next, we analysed the transgenic line carrying the snake *HoxD* cluster and found that, in this case, expression of *Hoxd4* in the trunk was not dorsally restricted as observed in all other vertebrate BACs assayed thus far. The expression pattern in the main body axis was in fact reminiscent of the endogenous mouse *Hoxd4* expression, with equally strong signals in both the neural tube and mesodermal derivatives (*Figure 2B*). Therefore, in contrast to other vertebrate species, enhancers located within the snake cluster appear sufficient to drive *Hoxd* gene expression in the ventral mesoderm. In order to assess if this increase in regulatory potential within the cluster was correlated with a reduction of long-range regulatory elements in the surrounding gene deserts we performed a comparative analysis of H3K27ac profiles between snake and mouse trunk tissues dissected from similar body parts. A global assessment of the profiles suggested that there were relatively less enriched sequences outside of the snake *HoxD* cluster than outside its mouse counterpart (*Figure 2D*, bottom). In order to be able to directly compare the ChIP-seq datasets in mouse and snake, we identified 27 DNA regions conserved between the two species and located within the telomeric desert and scored their enrichments with acetylation of H3K27. While 40% of these conserved sequences were acetylated in the mouse sample, only 22% of them were significantly decorated by this chromatin mark in the snake tissue (*Figure 2D*, right). Overall, these results indicate that the enhancers required to control snake *Hoxd* gene expression in the trunk mesoderm are, at least for the most part positioned within the cluster rather than in the telomeric gene desert.

These DNA segments acetylated in snakes were found clustered in two regions of the gene desert as demonstrated by peak calling, whereas the mouse acetylated DNA regions span a larger portion of the gene desert (*Figure 2D*). Of note, one of the acetylated peaks in the mouse was scored over a region conserved in mammals, birds and amphibians, but not in snakes (*Figure 2—figure supplement 3A and B*). To confirm the enhancer activity of this sequence (MSS), we cloned the mouse version upstream of a *LacZ* reporter gene. As expected, MSS was able to drive expression in

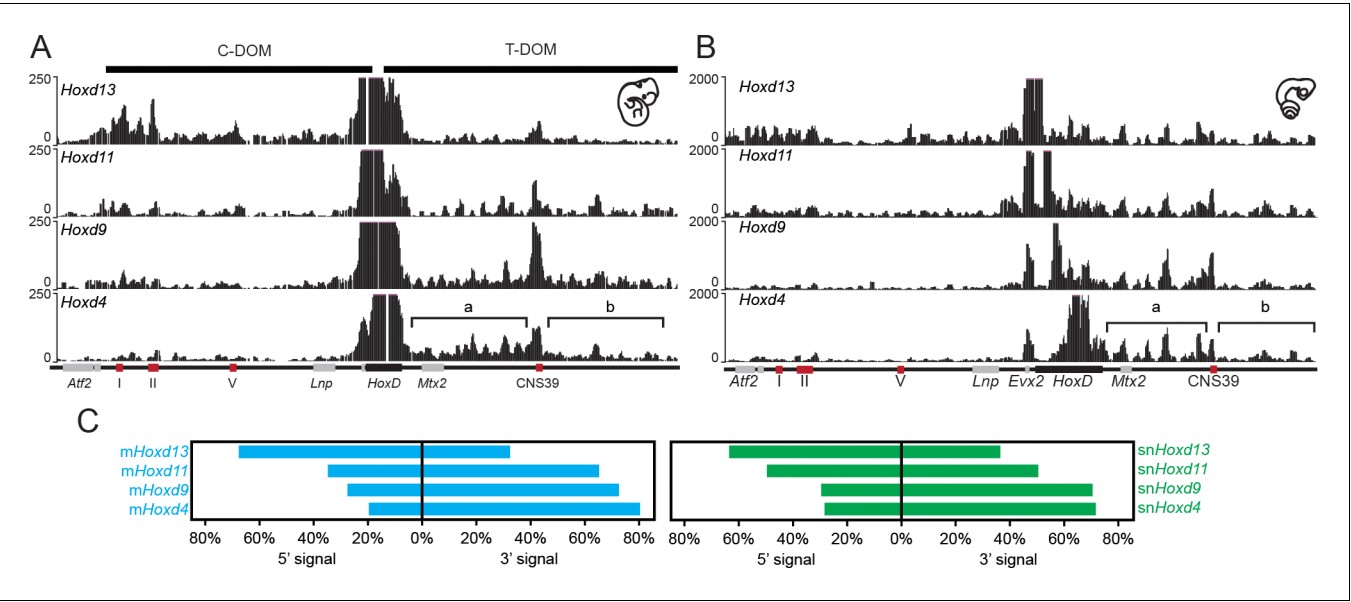

**Figure 3.** 4C-seq bimodal interaction profiles for *Hoxd* genes in mouse and snake embryos. (**A**) The four tracks show the interaction profiles established by either the *Hoxd13, Hoxd11, Hoxd9* or the *Hoxd4* viewpoints in E11.5 total mouse embryo. While *Hoxd13* mostly interacts with the centromeric landscape (left), *Hoxd4* contacts preferentially the telomeric landscape. Both *Hoxd11* and *Hoxd9* show intermediate profiles. The centromeric (C-DOM) and telomeric (T-DOM) TADs are represented as black boxes on top of the profiles. (**B**) The four tracks show the snake orthologous series of genes used as baits on 2.5 dpo corn snake whole embryos. The same general interaction profiles are observed. Brackets indicate the location of the two telomeric sub-TADs:' 'a' and 'b'. Under the profiles the *HoxD* locus is represented. The black rectangle is the *HoxD* cluster, grey boxes are neighbouring genes and red boxes represent known constitutive contacts in the mouse that are conserved in snakes. (**C**) Graphical representation of the percentage of interactions either in 5' or in 3' of the gene clusters, calculated for the different viewpoints for the mouse (blue) or the snake (green).

The following figure supplement is available for figure 3:

**Figure supplement 1.** 4C-seq in the telomeric gene desert of mouse and snake tissue.

the trunk mesoderm from the forelimb to more posterior parts of the embryo (*Figure 2—figure supplement 3C*).

## Bimodal regulation in the snake *HoxD* regulatory locus

At the mouse *HoxD* locus, a bimodal regulatory strategy associated with particular chromatin conformations was reported to control *Hox* gene expression in a variety of organs and structures. Such global controls involve separate sets of target *Hoxd* genes, which are thus re-activated after the major body axis is laid down (*Andrey et al., 2013*; *Spitz et al., 2001*). Most of these structures, however, are either missing in snakes, such as the limbs or the intestinal cecum or whenever present, they are nevertheless substantially different from their mammalian counterparts. Because mouse *Hoxd* genes contact such remote enhancer sequences *via* long-range interactions included within two opposite TADs (*Montavon et al., 2011*), we set out to see whether such a bimodal type of regulatory topology would also exist in snakes, even in the absence of many of the related functionalities. We thus used whole mouse and snake embryos of similar size to characterize the interaction profile of *Hoxd* genes with their surrounding regulatory landscapes.

We used the 4C-seq version of chromosome conformation capture (*Dekker et al., 2002*; *de Laat and Dekker, 2012*) with four different *Hoxd* genes as viewpoints to assess their potential interaction tropism with either one of the flanking gene deserts. As observed in the mouse, significant interactions between the snake viewpoints and the centromeric TAD were observed, mostly when the *Hoxd13* bait was used and, to a lower extent, with *Hoxd11* (*Figure 3A,B*, compare tracks 1 and 2). In both cases however, substantial contacts were also observed with the opposite, telomeric TAD. These latter interactions increased when the snake *Hoxd9* and *Hoxd4* baits were used, whereas at the same time, interactions with the centromeric landscape almost disappeared (*Figure 3A,B*, tracks

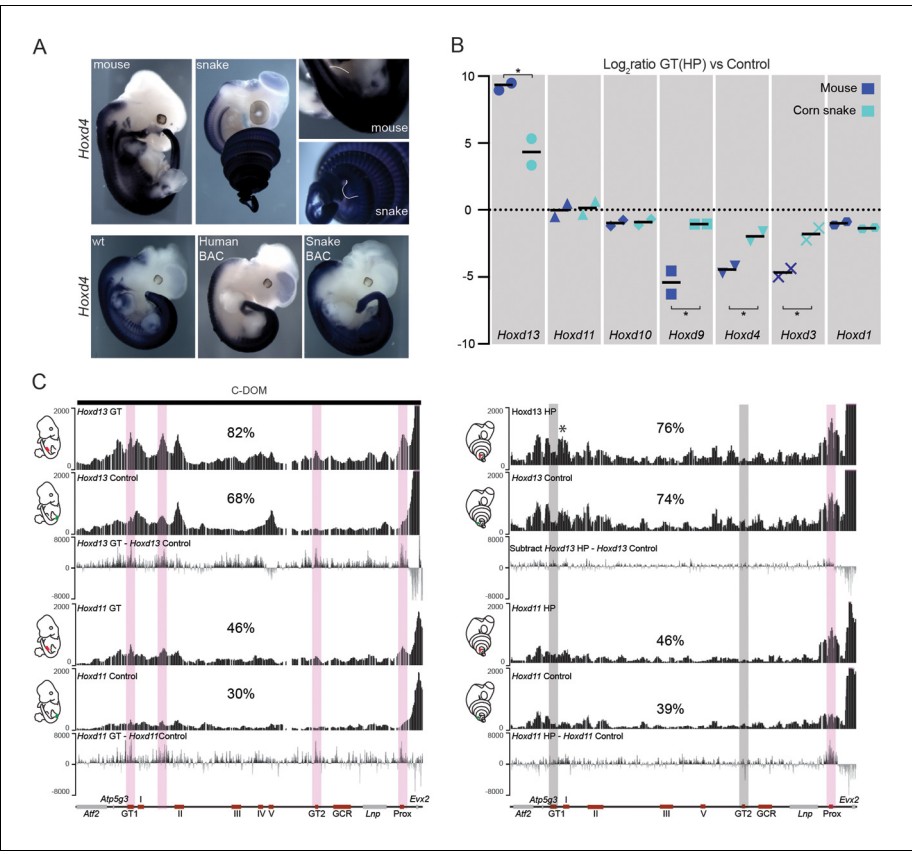

**Figure 4.** Regulation of mouse and corn snake *Hoxd* genes in developing genitals. (**A**) Endogenous *Hoxd4* expression both in a E12.5 control mouse embryo and in a 8.5 dpo corn snake embryo. Higher magnifications of the cloacal regions are shown on the right, with the positions of the GT and HP delineated in white. Below are in situ hybridization of either control or E11.5 embryos transgenic for the human and snake BAC clones using species-specific probes. (**B**) Quantifications of *Hoxd13*, *Hoxd11*, *Hoxd10*, *Hoxd9*, *Hoxd4*, *Hoxd3* and *Hoxd1* transcript levels either in mouse E12.5 GT (n=2) or in snake 4.5 dpo HP (n=2) by RT-qPCR. The $\log_2$ ratios were calculated between genital and control trunk tissue expression values. *Hoxd13* (P = 0.0378), *Hoxd9* (P = 0.0375), *Hoxd4* (P = 0.0298) and *Hoxd3* (P = 0.0342) $\log_2$ ratios are significantly different between mouse and corn snake while *Hoxd11* (P = 0.8303) and *Hoxd10* (P = 0.8539) values are not (*P < 0.05; unpaired two-tailed *t*-test). Bars indicate the average. (**C**) Smoothed 4C-seq mapping using mouse and snake *Hoxd13* and *Hoxd11* as viewpoints and GT (mouse) and HP (snake) as samples along with a control sample (left). The BamCompare subtract function was used for each viewpoint to compare sequence coverage in GT/HP *versus* control tissues. Genes are represented by grey rectangles and previously characterized mouse limb or GT enhancers are represented by red boxes below. The vertical shaded zones in pink represent sequences that displayed increased read coverage in GT versus control tissue, whereas the grey zones point to sequences showing increased contact in mouse but not in snake genitals. The percentages show the relative amount of interactions over this particular landscape, calculated as in *Figure 3*. The centromeric TAD C-DOM is represented by a black rectangle above the mouse 4C profile. An asterisk highlights strong contacts of *Hoxd13* with Island I in the snake.
The following figure supplement is available for figure 4:

**Figure supplement 1.** Interspecies comparison of the regulatory potential associated with the *HoxD* cluster.

3 and 4). Therefore, as previously reported in the case of mouse tissues and ES cells, genes located at different relative positions within the *HoxD* gene cluster show distinct interaction tropisms. 5'-located genes such as *Hoxd13* interact primarily with the 5'-located gene desert, whereas genes located in a more 3' part of the cluster, such as *Hoxd4*, interact mostly with the 3'-located gene desert (*Figure 3C*).

Interestingly, while the general tendency in the bimodal distribution of interactions was thus comparable between mouse and snake full embryos, the pattern of contacts presented important differences between the two species. In the mouse developing proximal limb for instance, *Hoxd11* and *Hoxd9* preferentially interact with the telomeric sub-TAD referred to as region 'b' rather than the sub-TAD 'a' (*Figure 3—figure supplement 1A and C*) (*Andrey et al., 2013*). Mouse region 'b' thus likely contains important proximal limb regulatory sequences. In whole embryo tissue, we found that the contacts were rather equally distributed between the two regions 'a' and 'b' (*Figure 3—figure supplement 1A and C*). In contrast, *Hoxd4* preferentially interacted with the sub-TAD 'a' (*Figure 3A*; brackets), similar to *Hoxd1* in proximal limbs (*Andrey et al., 2013*). In snake embryonic cells, however, *Hoxd9* and to a much lesser extent *Hoxd11*, displayed an interaction pattern related to that of *Hoxd4* with contacts enriched within the sub-TAD 'a', suggesting the absence of strong regulatory controls located in region 'b' of the snake gene desert (*Figure 3—figure supplement 1B and D*).

## Divergent evolution of genital bud-specific regulatory sequences

In the mouse, strong contacts between *Hoxd13* and its flanking regulatory landscape were associated with its function during the development of both digits and external genitals (*Lonfat et al., 2014*; *Montavon et al., 2011*). As snakes lack digits, we investigated whether the conservation of this particular chromatin domain was related to the existence of external genital organs. Male snakes display hemipenes (HP), resulting from symmetrical genital buds during development. As it was proposed that the genitals of mammals have a different embryonic origin than those of other amniotes such as squamates (*Tschopp et al., 2014*), the existence of the same global regulation in snakes was unclear.

In situ hybridization revealed that 3'-located genes such as *Hoxd4* are expressed in the snake HP (*Figure 4A*, top), in contrast to the mouse where neither *Hoxd3* nor *Hoxd4* are transcribed in this structure (*Lonfat et al., 2014*). This difference was also scored when *Hox* gene expression was analysed in the various BAC transgenic lines. While the human, mouse, chicken and zebrafish *Hoxd* genes were expressed mostly along the main body axis and transcribed neither in the limbs, nor in the external genitals, in agreement with previous results (see above and (*Lonfat and Duboule, 2015*) (*Figure 4A*, bottom and *Figure 4—figure supplement 1*), the snake BAC expressed the *Hoxd11* to *Hoxd4* genes in the developing limbs and genital bud (*Figure 4A* and *Figure 4—figure supplement 1*). This likely reflects a lack of repression of these genes into such structures, rather than the presence of limb and genital enhancers located inside the cluster. We investigated this issue by RT-qPCR in the incipient genitals of both mouse and snake, using as a control a region of the trunk located at the exact same anterior-posterior level. In such conditions, while the mouse *Hoxd9* to *Hoxd3* genes were expressed at much higher levels in the trunk when compared to the genitals, the steady state levels of snake mRNAs were nearly the same in both tissues (*Figure 4B*). Therefore, these results suggested that the snake *HoxD* cluster lacks the sequences necessary to prevent *Hox* gene expression from the trunk lateral mesoderm, a critical factor to properly develop limbs and genitals in mouse (see discussion).

## Enhancer evolution

To further explore this difference in *Hox* gene regulation between murine and snake external genitalia, we assessed whether *Hoxd* gene regulation in the developing genitals also relied upon enhancer sequences located in the regulatory landscape upstream of *Hoxd13*. We performed 4C-seq analyses using embryonic mouse GT and snake HP tissues at comparable stages, as well as control trunk tissues. As expected, both the mouse *Hoxd11* and *Hoxd13* genes showed more interactions with the centromeric gene desert in the GT material than in control trunk material (*Figure 4C*). In contrast, the snake *Hoxd13* and *Hoxd11* interaction profiles were not significantly different, when either HP or the control samples were used. We searched the snake gene desert for the presence of the GT1 and GT2 sequences, two DNA segments described in the mouse counterpart to specifically interact with *Hoxd13* in the developing GT (*Lonfat et al., 2014*) (*Figure 4C*, bottom left) and could identify them (*Figure 4C*, bottom right). However, even though these sequences are well conserved in the snake, they did not significantly increase their interactions with *Hoxd13* during the development of the snake genitals (*Figure 4C*, right). In fact, the comparative analysis of the snake 4C-seq data revealed

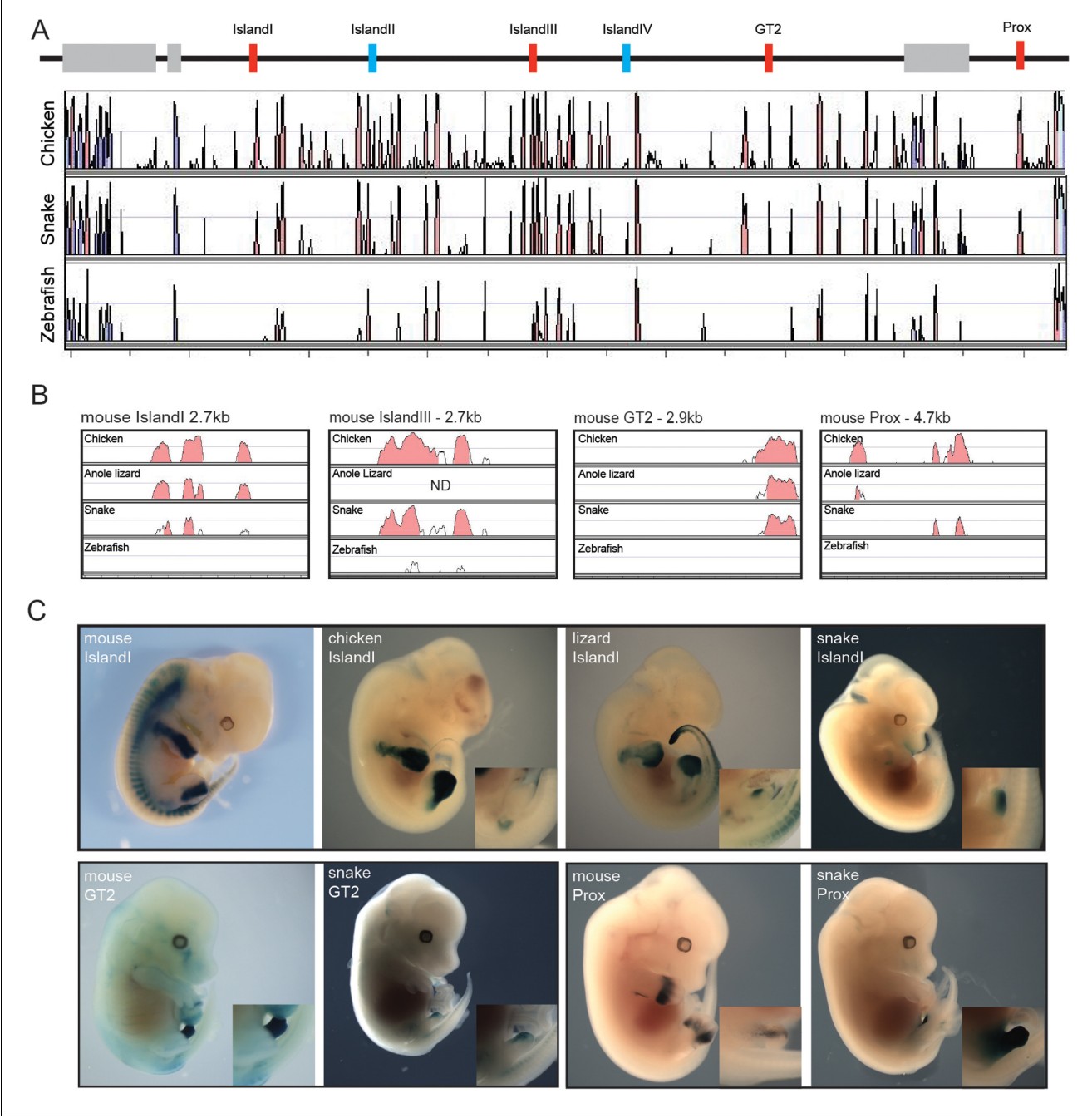

**Figure 5.** Enhancer activity of mouse limb and GT enhancers. (**A**) Conservation plot over the 5' gene desert (centromeric in the mouse) using mouse as reference sequence. Peaks represent a conservation higher than 50%. The alignment was made with the mVista program using sequences from mouse, chicken, corn snake and zebrafish. Genes are represented by grey boxes. The various mouse limb and/or GT enhancers conserved from mammals to chicken are represented by red boxes, whereas mouse limb enhancers either poorly or not conserved at all in chicken are in blue. (**B**) Conservation plots of selected mouse limb and GT enhancers using the mouse sequence as a reference. Coloured peaks represent a conservation of above 75%. (**C**) Enhancer activities of the mouse, chicken, lizard and snake Island I (E12.5) (top), and mouse and snake enhancer activities of the GT2 (E14.5) and Prox (E12.5) sequences (bottom), in transgenic mouse foetuses. Magnifications of the genital region are included.

that, if anything, only the Prox sequence, a known mouse GT and limb enhancer (*Gonzalez et al., 2007*) appeared to gain interactions in the snake HP sample, when compared to control tissue.

Therefore, in addition to the fact that snakes are limbless and that their external genitalia may derive from a different embryonic origin (*Tschopp et al., 2014*), our results pointed to distinct

modalities in the implementation of global gene regulation at the *HoxD* locus. Consequently, we searched for the presence within the snake centromeric TAD of the digit- and genital bud-specific enhancers previously identified in mammals (*Lonfat et al., 2014*; *Montavon et al., 2011*). Surprisingly, all these murine enhancers conserved within mammals and up to birds showed some level of conservation in snakes (*Figure 5A*). To try and assess the functional potential of these sequences, we isolated the snake sequences Prox, GT2 and Island I (*Figure 5A,B*) and used them separately in a mouse transgenic enhancer assay. The mouse counterpart of the Prox sequence displayed activity in both the developing digits and GT. The mouse GT2 sequence is a genital only-specific enhancer and the mouse Island I sequence displays limb-only enhancer specificity when placed upstream of a *lacZ* reporter in a transgenic assay (*Montavon et al., 2011*; *Gonzalez et al., 2007*; *Lonfat et al., 2014*) (*Figure 5C*, top).

The snake Prox element elicited a robust *lacZ* staining in the developing mouse GT. Interestingly however, and in contrast to the mouse sequence, staining was not detected in the growing limb buds, indicating that the snake Prox sequence had lost its potential to drive transcription in digits (*Figure 5C*, bottom). Likewise, the snake GT2 sequence was able to drive reporter gene expression in the mouse GT. However, the level of *lacZ* staining obtained was consistently weaker than with the mouse GT2 sequence (*Figure 5C*, bottom), perhaps related to the weak (if any) contact observed by 4C between the snake *Hoxd13* gene and this sequence (*Figure 4C*). Therefore, in the two cases where a mouse sequence displayed an enhancer potential for the developing GT, the cognate snake sequence appeared to share part or all of this potential (*Figure 5C*, bottom).

We then turned to the snake Island I, a mouse limb-specific enhancer sequence and, noteworthy, the snake version was able to drive reporter gene expression in the mouse GT while unable to elicit any staining in the developing limb buds (*Figure 5C*, top). Therefore, in this case, the same regulatory sequence conserved between the mouse and the snake was interpreted either as a limb- or as a genital-specific enhancer by the mouse, when introduced as transgenes. To further evaluate this striking change in enhancer potential, we cloned and investigated the regulatory capacity of both the chicken and the green anole lizard Island I sequences. The chicken construct elicited staining in limbs but not in the GT, i.e. in a pattern reminiscent of the mouse rather than the snake Island I (*Figure 5C*, top). Noteworthy, while the lizard Island I also drove reporter gene expression in limbs, weak staining was scored in the GT, somehow displaying an intermediate enhancer specificity between the snakes and the other amniotes assayed (*Figure 5C*, top).

## Discussion

Proper sequential *Hox* gene activation in time and space during the elongation of the main body axis is critical for the correct patterning of the axial skeleton (*Deschamps and van Nes, 2005*). While the underlying regulatory mechanisms remain to be fully understood (see e.g. (*Gaunt, 2015*)), they likely involve control sequences located both inside and outside of the *Hox* gene clusters (*Tschopp et al., 2009*; *Tschopp and Duboule, 2011*) as well as concurrent epigenetic modifications (*Soshnikova and Duboule, 2009*).

Transgenic approaches have identified several *cis*-regulatory elements, which could reproduce, for the most part, a *Hox*-like expression pattern in the main body axis and which mapped close to their target gene(s) (for examples, see (*Bel-Vialar et al., 2002*; *Brend et al., 2003*; *Charité et al., 1995*; *Kwan et al., 2001*; *Oosterveen et al., 2003*; *Sharpe et al., 1998*)). However, our various transgenic mouse lines containing *HoxD* clusters from different vertebrate species showed that, in most vertebrates, regulatory sequences located in the gene deserts flanking the *HoxD* cluster are necessary for proper expression in the embryonic trunk mesoderm. Indeed, when using the expression of *Hoxd4* as a read-out of the regulatory potential contained either in the mouse, the human, the chicken or the zebrafish clusters, ventral mesodermal expression was not detected. This result is at odds with previous reports describing the presence of mesodermal enhancers on a short mouse transgene derived from this locus (*Zhang et al., 1997*; *Morrison et al., 1997*). Similarly, we found *Hoxd3* expression to be absent from the ventral mesoderm in mouse embryos that lack the telomeric gene desert while previous work has reported that a small single-copy construct containing the *Hoxd3* locus was able to drive reporter gene expression in this tissue (*Tschopp et al., 2012*).

These discrepancies may derive from a qualitative aspect whereby a short transgene may reveal the potentialities absent from a more complex BAC environment, such as in the case of *Hoxd3*. They

may also reflect quantitative differences, for example when comparing (close to) single copy BAC integrations with a large number of head to tail integrations of a shorter transgene, as for *Hoxd4*. In any case, a different result was obtained when the snake *HoxD* BAC was used since, unlike the other vertebrate clusters tested, the snake transgene appeared to contain the regulatory elements necessary for expression in the trunk mesoderm. The H3K27 acetylation profiles in mouse and corn snake trunks supported this result for only few acetylation peaks were scored outside of the snake cluster compared to the mouse. In an evolutionary context, it is possible that in snakes, long-range mesodermal enhancers were progressively complemented by local enhancers to regulate *Hoxd* gene expression in the developing body axis, perhaps to provide an increased fine-tuned control in the expression balance between single *Hoxd* genes during embryonic development.

In the absence of any genetic approach to study snake development, it is difficult to evaluate the functional relevance of this difference in regulation. It is however worth noting that the snake *HoxD* cluster BAC was the only transgenic configuration where expression was strong in the limbs and external genitals, whereas BACs derived from animals with limbs did not elicit an expression of *Hoxd* genes into the transgenic limbs, even though the endogenous genes were strongly expressed there. Also, isolated mouse mesodermal enhancers were often described to activate reporter gene expression in secondary structures such as limbs (*Charité et al., 1995*; *Kwan et al., 2001*; *Sharpe et al., 1998*; *Renucci et al., 1992*). In tetrapods, this apparent paradox may reflect the necessity for a highly specific type of regulation to control both *Hoxa* and *Hoxd* genes in specific domains of the growing limb buds (*Andrey et al., 2013*; *Berlivet et al., 2013*; *Woltering et al., 2014*). Implementing such global limb regulations may require previous regulatory inputs to be terminated. Our results using the corn snake BAC suggest that, in the absence of limbs, this negative control may have been lost in the course of evolution, leading to the maintenance of transcription in all mesoderm derivatives. Whether or not this increased 'mesoderm potential' present in the snake *HoxD* cluster may somehow relate to the presence of ventral mesodermal enhancer remains to be clarified.

*Hox* clusters of jawed vertebrates show high levels of chromatin compaction (*Noordermeer et al., 2011*; *Fabre et al., 2015*), a feature associated with the unusual level of gene packaging and organization, which occurred at the roots of the vertebrate lineage (*Duboule, 2007*) together with the emergence and generalisation of long-range regulations at these loci (*Darbellay and Duboule, 2016*). Squamate *Hox* clusters however seem to slightly deviate from this rule by having accumulated a large number of transposable elements (*Di-Poï et al., 2009*), a situation rarely found around genetic loci of developmental importance (*Simons et al., 2007*) due to the potential effects of such sequences to elicit genetic and morphological variations (*Kidwell and Lisch, 1997*; *Friedli and Trono, 2015*). While the presence of such repeated elements within and around the snake *HoxD* cluster may have been associated with differences in the location of enhancer sequences, the distribution of chromatin modifications did not point to any drastic regulatory reorganization of this gene cluster in snakes. Indeed, the H3K9me3 histone mark, which in some cases is associated with TEs such as LTRs, LINEs and SINEs (*Friedli and Trono, 2015*; *Mikkelsen et al., 2007*) was not found within the *HoxD* cluster itself. In addition, the analysis of other chromatin marks present in trunk tissue during the sequential activation of the gene cluster displayed distributions similar to those found in the mouse cluster.

These global similarities between mouse and snake in the structure of the *HoxD* cluster were also noticed when interaction profiles were considered. There again, the snake embryonic material displayed the bimodal distribution of contacts on both sides of the gene cluster, as expected either from several studies using specific mouse samples (*Lonfat et al., 2014*; *Andrey et al., 2013*), or from the full embryonic material used in this work. In all cases, most *Hoxd* genes tend to naturally interact within the 'telomeric' (3'-located) TAD, whereas *Hoxd13* was strongly associated with the 'centromeric' (5'-located) TAD. However, a significant difference was scored in the interaction profiles of *Hoxd9*, which displayed more contacts further away of region CNS39 in the mouse than in the snake sample. As this regulatory region is particularly active in proximal limbs, the lower frequency of contacts observed in snake embryos was not unexpected. This observation indicates that, while the general bimodal TAD regulatory structure is conserved at the *HoxD* locus, differences between vertebrate species may exist either in the extent, or in the internal organization of interactions within the TADs.

This latter point was of particular interest as the mouse centromeric TAD was initially defined due the presence of many limb and genital-specific enhancers (*Montavon et al., 2011*; *Lonfat et al., 2014*), which were not necessarily expected to be conserved in the snake orthologous landscape due to both the absence of limbs and the presumably distinct origin of snake external genitalia following the relative shift of the cloaca over the course of evolution (*Tschopp et al., 2014*). However, tetrapod limb-specific enhancers are often conserved in snakes and a significant overlap between limb and genital *cis*-regulatory mechanisms was recently reported (*Infante et al., 2015*), in agreement with the similarities between the molecular mechanisms employed to generate the two structures (*Kondo et al., 1997*; *Cohn, 2011*; *Yamada et al., 2006*). Indeed when a mouse limb- and genital-specific *Tbx4* enhancer sequence was isolated from snake, it could only recapitulate genital expression, thus having lost the limb-specific regulatory potential (*Infante et al., 2015*).

Consistent with this observation, we find that the snake counterpart of the mouse limb and genital enhancer Prox (*Gonzalez et al., 2007*) has lost its limb regulatory potential, while keeping a strong capacity to drive expression in the developing genitalia. This suggests that the mouse Prox consists of two regulatory modules, while the snake Prox has kept the genital specificity only. More strikingly however, Island I, which in the mouse is a limb-only specific enhancer (*Montavon et al., 2011*), switched its regulatory capacity in the snake to become a genital-only specific enhancer, accompanied by elevated enhancer-promoter interactions as assayed by 4C (*Figure 4C* – asterisk). On the other hand, the chicken Island I revealed enhancer specificities almost identical to those of the mouse sequence despite the fact that birds are more closely related to squamates than to mammals, suggesting that Island I had a limb-only activity at the time of divergence between mammals and reptiles/birds. Interestingly, the lizard sequence displayed a weak activity in the genital bud, in addition to the limb, indicating that the co-option to a genital function likely occurred in the squamate clade and thus preceded limb loss in snakes.

External genitalia of squamates, unlike that of mammals, were proposed to have the same embryonic origin as limbs (*Tschopp et al., 2014*). Although, this could bias our mouse-based transgenic analysis, the snake Prox sequence could drive reporter gene expression specifically in the mouse GT and not in the limb, making it unlikely that differences in embryonic origin could interfere with this particular analysis. While the generation of the same transgenics in snakes would be necessary to fully rule out this bias, such experiments are not feasible for the moment.

Altogether our results show that, while the general bimodal regulatory strategy is conserved, some profound differences in the regulation are scored at the *HoxD* locus between two species displaying strikingly distinct morphologies. It is as yet unclear if such changes were causative of the extensive morphological changes that snakes experienced over the course of evolution, or whether they are merely consequential. It nevertheless indicates that vertebrates with extreme variations in those systems known to be under the control of *Hox* genes (vertebral number and identities, limbs, genitals) rely upon the same general regulatory architecture and principles at *Hox* loci. In this view, vertebrate morphological evolution was accompanied by changes in *Hox* gene regulation, yet such variations were constrained within the general regulatory framework found at these loci. This may reflect selective pressures that impose essential basic properties to vertebrate body plans, while other more subtle morphological specificities, less likely to result in adverse effects, may arise in the course of evolution.

## Materials and methods

### Animal maintenance

Maintenance of, and experiments on animals were approved by the Geneva Canton ethical regulation authority (authorization 1008/3421/1R to M.C.M. and GE/81/14 to D.D.) and performed according to Swiss law.

### BAC library construction, screening and sequencing

A *Pantherophis guttattus* BAC library containing 55'296 clones was constructed from liver tissue of one single individual (Amplicon Express). Degenerate primers were designed in DNA regions conserved between mammallian and bird species within the DNA interval spanning from the *Atf2* gene to the CNS65 region (mouse chr2:73653618–75292344 in mm9). The amplified DNA fragments

ranging from 400 bp to 1 kb (kilobases) were cloned into a PGEMTeasy vector and labelled with DIG-High Prime (Roche, Switzerland) to screen filters provided by the company. The lengths and positions of positive BACs were evaluated by PCR and BAC end sequencing. 13 BAC clones were selected for sequencing at BGI (Beijing Genomics Institute). Sequencing was performed on 500 bp and 2 kb-large insert libraries using the Illumina HiSeq2000 and assembly was done using SOAP denovo. The resulting sequence is deposited in GenBank under accession number KU866087.

## Sequence analysis and annotation

Exons of the corn snake *Hox* genes were identified using GENSCAN (http://genes.mit.edu/GEN-SCAN.html) (*Burge and Karlin, 1997*) and sequence comparison against *Hox* coding sequences of closely related species. Conservation of non-coding elements was assessed by the use of the mVista software (http://genome.lbl.gov/vista/mvista/submit.shtml) with default parameters. Information about different vertebrate *Hox* clusters and sizes of gene deserts were taken from the UCSC genome browser (http://genome.ucsc.edu) and the ncbi genome database (http://www.ncbi.nlm.nih.gov/genome/). Transposable elements were identified using RepeatMasker (http://www.repeatmasker.org/) and the Repbase vertebrate repeat library combined with a *de novo* corn snake repeat library described in *Ullate-Agote et al. (2014)*.

## ChIP-sequencing

Snake forebrain, anterior trunk and posterior trunk samples as well as mouse anterior trunk samples were dissected and fixed in 1% formaldehyde for 10 min. For each ChIP-seq experiment approximately 100 ng of tissue were used and processed according to (*Lee et al., 2006*) or the ChIP-IT High Sensitivity (Active motif) specifications. H3K27me3 antibody (Millipore, 17–622), H3K27ac antibody (Abcam ab4729) and H3K9me3 (Abcam ab8898) were used. Sequencing was performed using 100 bp single-end reads in the Illumina HiSeq system according to manufacturer's instructions. The reads obtained from the sequencing were mapped to ENSEMBL Mouse assembly NCBIM37 (mm9) or to the corn snake scaffold using the HTSstation mapping pipeline (http://htsstation.epfl.ch) (*David et al., 2014*). All ChIP-seq mappings were normalized to total input chromatin using the bamCompare software from the deepTools Galaxy web server (http://deeptools.ie-freiburg.mpg.de) (*Ramirez et al., 2014*). Peak calling was done using MACS (*Zhang et al., 2008*).

## Mouse stocks

The *HoxD*[Del(AttP-SB3)] (aka Del(*AttP-SB3*)), *HoxD*[Del(AttP-SB2)] (aka Del(*AttP-SB2*)) and *HoxD*[Del(SB2-65)] (aka Del(*SB2-65*)) mutant alleles were generated through TAMERE (*Hérault et al., 1998*) and have been described elsewhere (*Andrey et al., 2013*). The *HoxD*[Del(Atf2-Nsi)](aka Del(*Atf2-Nsi*)) allele was also produced by TAMERE and described in (*Montavon et al., 2011*). The *HoxD*[Del(1–13)d11lacZ] (aka Del(*1–13*)d11lacZ allele, previously referred to as *Del9* in *Zákány et al. (2001)* was obtained by loxP/Cre mediated recombination in ES cells. The *HoxD*[inv(AttP-CD44)] (aka Inv(*AttP-CD44*)) was generated using STRING (*Spitz et al., 2005*). Telomeric desert deletions: The *HoxD*[Del(SB2-SB3)](aka Del(*SB2-SB3*)) and Del(*65-SB3*) alleles were generated by TAMERE. All telomeric deletions analyzed were trans-heterozygotes over the *HoxD*[Del(1–13)d11lacZ] balancer allele. The mouse (RP23-400H17) BAC has been previously described in (*Spitz et al., 2001*). The human (CTD-2086D13), chicken (CH261-92D10), snake (Eg-32P1 custom library) and zebrafish (77g24) BACs were all recombined to introduce a *PIScel* site in the vector using EL250 cells (*Lee et al., 2001*). The snake and chicken BACs were further shortened to remove all sequences flanking the *HoxD* cluster. The final genomic coordinates of the chicken BAC used for transgenesis were chr7:17361344–17440245 (galGal3). The snake BAC, which initially extended 74 kb downstream of *Hoxd1* was reduced to contain only 300 bp 3' of *Hoxd1*. Successful recombination was confirmed by PCR, restriction digest and BAC end sequencing. Prior to injection, BACs were isolated using the Nucleobond Maxiprep kit, linearized with *PISceI*, purified by phenol/chloroform extraction and dialyzed against microinjection buffer. The linearized BACs were then injected into fertilized mouse oocytes. After having obtained transgenic lines, BAC integrity was confirmed by PCR using primers specific for *Hox* genes of the appropriate species.

**Table 1.** List of primers used to clone the probes for in situ hybridization.

| Hs*Hoxd13* | GGTCCAGGTTGGCCACAGAC |
|---|---|
| | GTCACTCTACTGATTGCAGC |
| Hs*Hoxd11* | TTGAGAGCTCCAGGAAGCGC |
| | TTCAGTTGCATGGGTTCTGG |
| Hs*Hoxd9* | CCAATTCCAAGAATGAAGGC |
| | ACATTTACAACTGGTCCTCG |
| Hs*Hoxd4* | CAACTCAGAGGCGAGTTCAC |
| | TCAAGTAGCTTGCTATGGCA |
| Dr*Hoxd13* | ATGATGGTTTCCAGATATGC |
| | TGGTGACAGCTGCCCAATCA |
| Dr*Hoxd11* | GAGCCGCTGTTCTTTTCTTC |
| | GTCCTATCCGCACGCATATG |
| Dr*Hoxd10* | CCACCTTTGCCTTCTCTGTG |
| | TCCAAAATGTCCTTTCCCAAC |
| Dr*Hoxd9* | TTACTTGGGTCAAGTTGTTG |
| | GTGAAGGCAGCAAAAATACT |
| Pg*Hoxd13* | GCGCTTCTGATCATGTTTGC |
| | ATAGCTAAACATATAGGCAC |
| Pg*Hoxd11* | CCTAGAGGTTAATATGACTCC |
| | CCCATTTAGGCTCCTAGG |
| Pg*Hoxd10* | CCGAGAACTGACTGCTAATC |
| | CAGAATTTATTGCATTATAC |
| Pg*Hoxd9* | AGGAGAGTAACACTTTGAGG |
| | CCTCTCTGACATGAGTCTTG |
| Pg*Hoxd4* | CGGATTTGACCACTTTATAG |
| | AACAATATCACCAACACATG |

**Table 2.** List of primers used for 4C-seq amplifications with snake tissues.

| Pg*Hoxd13* DpnII | AATGATACGGCGACCACCGAACACTCTTTCCCTACACGACGCTCTTCCGATCTGGAAAAGGTTGTTAATCAGG |
|---|---|
| Pg*Hoxd13* NlaIII | CAAGCAGAAGACGGCATACGACTGCCCTTCTTCAAAGAGAC |
| Pg*Hoxd11* NlaIII | CAAGCAGAAGACGGCATACGAGCCGCAGTTGTCCAAGTTAC |
| Pg*Hoxd11* DpnII | AATGATACGGCGACCACCGAACACTCTTTCCCTACACGACGCTCTTCCGATCTTCCTCCTTGAGAGGGAATCC |
| Pg*Hoxd9* NlaIII | CAAGCAGAAGACGGCATACGAAAGAATCCCCATCCTAGTCC |
| Pg*Hoxd9* DpnII | AATGATACGGCGACCACCGAACACTCTTTCCCTACACGACGCTCTTCCGATCTTGTAATCGTAATCAGCATAG |
| Pg*Hoxd4* DpnII | AATGATACGGCGACCACCGAACACTCTTTCCCTACACGACGCTCTTCCGATCTCACTTCATCCTTCGGTTCTG |
| Pg*Hoxd4*NlaIII | CAAGCAGAAGACGGCATACGATAAACAATGAAGTGAAACGG |

## In situ hybridization and probe design

In situ hybridization was performed as previously described (*Woltering et al., 2009*) with a hybridization temperature of 68°C and 1.3x SSC concentration in the hybridization mix. Post-hybridization washes were performed using 2x SSC concentration for four times 30 min. All probes designed for in situ hybridization of BAC transgenic embryos were tested for cross-reactivity by conducting in situ hybridization on control mouse embryos. To produce human, chicken, snake and zebrafish riboprobes, DNA fragments were amplified from BAC DNA that comprised either the first exon or the 3'UTR of the *Hox* genes (see *Table 1*). After ligation with the PGEMTEasy vector (Promega), probes were synthetized using DIG RNA labeling mix (Roche) and purified with the QIAGEN RNeasy mini kit. *Hoxd11*, *Hoxd9* and *Hoxd4* mouse probes were previously described (*Gérard et al., 1996*; *Zappavigna et al., 1991*; *Featherstone et al., 1988*).

**Table 3.** List of snake and mouse primers used for qPCR.

| Pg*Hoxd13* | ACGAGACCTACATCTCCATG |
|---|---|
| | TTGGTGTAAGGCACTCGCTTC |
| Pg*Hoxd11* | TCCGAAAAGCCAGAGTTCAG |
| | ATCTGGTACTTGGTGTAAGG |
| Pg*Hoxd10* | CGTCTCCAGCCCAGAAAGC |
| | GGTTGGAGTATCAGACTTGG |
| Pg*Hoxd9* | AGGAAAAAGAGGAGCAGCAG |
| | TGGAGCGAGCATGAATCCAG |
| Pg*Hoxd4* | GAAAGTCCACGTTAACTCTG |
| | GACTTGCTGCCTGGTATAAG |
| Pg*Hoxd3* | AGGTATCCAGCTCGCTTACC |
| | GCGGACTCTTGTCTTCACAG |
| Pg*Hoxd1* | AAAGTCAAGAGGAACGCACC |
| | ACTGGAAGACCCACAAGCTG |
| PgHmbs | ATTGGGACCAGCTCACTTCG |
| | CCTCCTTCTCGTCCAGCTTC |
| Mm*Hoxd13* | GAAATCATCCTTTCCAGGAGATG |
| | CGCCGCTTGTCCTTGTTAATG |
| Mm*Hoxd11* | AAGAGCGGCGGCACAGTG |
| | TTGAGCATCCGAGAGAGTTGG |
| Mm*Hoxd10* | AGGAGCCCACTAAAGTCTCC |
| | CAGACTTGATTTCCTCTTTGC |
| Mm*Hoxd9* | GACCCAAACAACCCTGCAG |
| | TTCAGAATCCTGGCCACCTC |
| Mm*Hoxd4* | TGCACGTGAATTCGGTGAAC |
| | GTGAGCGATTTCAATCCGACG |
| Mm*Hoxd3* | AAGCAGAAGAACAGCTGTGC |
| | TAGCGGTTGAAGTGGAACTCC |
| Mm*Hoxd1* | GGCCCTTTCAGACTGTGTCC |
| | CATATTCGGACAGTTTGCTTTTC |
| MmHmbs | CGGCTTCTGCAGACACCAG |
| | CCCTCATCTTTGAGCCGTTTTC |

### Enhancer transgenesis and *lacZ* staining

Prospective enhancer sequences were obtained by either PCR or restriction digest and cloned upstream of a βglobin-*lacZ* construct into either PGEMTEasy, or SK Bluescript(-). Constructs were injected into mouse oocytes and embryos were harvested at E12.5 and E14.5. Beta-galactosidase staining was performed by fixing in 4% PFA for 30 min, washing in PBS/0.1% Tween and incubating in staining solution (1 mg/mL Xgal) overnight at 37°C. A minimum of three transgenics with consistent staining was obtained per construct. Mouse embryos transgenic for either the Island I/LacZ, the GT2/LacZ or the Prox/LacZ constructs had been obtained in previous studies (*Lonfat et al., 2014*; *Montavon et al., 2011*; *Gonzalez et al., 2007*).

### 4C-sequencing

One E11.5 whole mouse embryo, one 2.5 dpo (days post oviposition) whole snake embryo, ca. 30 mouse E13.5 and corn snake 8.5 dpo genital buds as well as mouse and snake trunk tissue dissected from comparable anterior-posterior levels were processed as previously described (*Noordermeer et al., 2011*). Mouse libraries were constructed by using NlaIII as the first restriction enzyme and DpnII (New England Biolabs) as the second restriction enzyme and the baits and inverse primers used for the *Hoxd4*, *Hoxd9*, *Hoxd11* and *Hoxd13* viewpoints were described in *Noordermeer et al. (2011)*. Snake libraries were constructed using DpnII as primary enzyme and NlaIII as secondary enzyme. For the *Hoxd4*, *Hoxd9*, *Hoxd11* and *Hoxd13* baits, the primers are listed in *Table 2*. All libraries were sequenced in the Illumina HiSeq system to generate 100 bp read length. The reads obtained were then demultiplexed, mapped and analysed using the HTSstation pipeline (http://htsstation.epfl.ch) (*David et al., 2014*). The global quantification of telomeric *versus* centromeric signals was calculated as in (*Andrey et al., 2013*). Signals mapping 5' of the cluster signal were quantified from the *Atf2* gene to 14 kb upstream of the *Evx2* gene and 3' signals were quantified starting from 5 kb downstream of *Hoxd1* to 46 kb downstream of the CNS65 enhancer sequence. Quantifications of contacts within regions 'a' and 'b' were calculated using the same coordinates as for the 3' signal calculation and by excluding the region chr2: 74964245–75004987 (mm9) in mouse and a comparable interval in snake so that the peak of interaction over the CNS39 region would not be accounted for. Comparison of signals between genital tissue and trunk control tissue was done by using bamCompare subtract function from the deepTools Galaxy web server (http://deeptools.ie-freiburg.mpg.de) (*Ramirez et al., 2014*).

### RT-qPCR

RNA from genitals and control trunk tissue was extracted from two E12.5 mouse embryos and from two 4.5 dpo corn snake embryos using the microRNeasy kit (QIAGEN). Biological replicate number was dependent on restricted availability of material and reduced variability of expression values between samples. cDNA was generated using the Promega GoScript reverse transcriptase according to manufacturer's instructions. qPCR was performed using SYBR select master mix (Applied Biosystems) using two technical replicates per biological sample. Primers used are listed in *Table 3*. The Hmbs gene expression was used for normalisation and $\log_2$ ratios were calculated between GT or HP expression values and trunk control tissue expression.

### Accession numbers

Raw and processed data of 4C-seq and ChIP-seq analysis are available in the Gene Expression Omnibus (GEO) repository under accession number GSE79048.

## Acknowledgements

We thank members of the Duboule laboratories for sharing material and discussions and Didier Trono for advices. We also thank Bénédicte Mascrez for help with mouse mutant stocks, Thomas Montavon for providing a control panel in *Figure 5C* as well as Athanasia Tzika for co-managing and Adrien Debry and Florent Montange for the caretaking of the corn snake colony. We thank the Geneva Genomics Platform and transgenic facilities (University of Geneva) as well as the Bioinformatics and Biostatistics Core (BBCF) and transgenic facilities of the Ecole Polytechnique Fédérale (EPFL) in Lausanne, for their assistance. This work was supported by funds from the Fundação para a

Ciencia e a Tecnologia (PTDB/BEX-BID/0899/2014) (to MM) and the University of Geneva, the EPFL, the Swiss National Research Fund (SNSF), the European Research Council (ERC) grants SystemsHox.ch and the Claraz Foundation (to DD).

## Additional information

### Funding

| Funder | Grant reference number | Author |
| --- | --- | --- |
| Fundação para a Ciência e a Tecnologia | PTDB/BEX-BID/0899/2014 | Moises Mallo |
| Schweizerischer Nationalfonds zur Förderung der Wissenschaftlichen Forschung | 310030B_138662 | Denis Duboule |
| European Research Council | 232790 | Denis Duboule |
| Claraz Foundation | | Denis Duboule |
| Université de Genève | | Denis Duboule |
| Instituto Federal de Educação, Ciência e Tecnologia do Espírito Santo | | Denis Duboule |

The funders had no role in study design, data collection and interpretation, or the decision to submit the work for publication.

### Author contributions

IG, Conception and design, Acquisition of data, Analysis and interpretation of data, Drafting or revising the article; SG, THNH, Acquisition of data, Analysis and interpretation of data; AN, Did all the transgenic animals for the first submission, Acquisition of data; JC, Produced transgenic animals for the revised version, Acquisition of data; FG, Acquisition of data, Contributed unpublished essential data or reagents; MCM, Drafting or revising the article, Contributed unpublished essential data or reagents; MM, Acquisition of data, Analysis and interpretation of data, Drafting or revising the article; DD, Conception and design, Analysis and interpretation of data, Drafting or revising the article

### Author ORCIDs
Moises Mallo, http://orcid.org/0000-0002-9744-0912
Denis Duboule, http://orcid.org/0000-0001-9961-2960

### Ethics

Animal experimentation: Maintenance of, and experiments on animals were approved by the Geneva Canton ethical regulation authority (authorization 1008/3421/1R to M.C.M. and GE/81/14 to D.D.) and performed according to Swiss law

## Additional files

### Major datasets

The following datasets were generated:

| Author(s) | Year | Dataset title | Dataset URL | Database, license, and accessibility information |
| --- | --- | --- | --- | --- |
| Isabel Guerreiro, Denis Duboule | 2016 | Reorganization of HoxD regulatory landscapes during the evolution of a snake-like body plan | http://www.ncbi.nlm.nih.gov/geo/query/acc.cgi?acc=GSE79048 | Publicly available at NCBI Gene Expression Omnibus (accession no: GSE79048) |

| Isabel Guerreiro, Denis Duboule | 2016 | CornSnake_HoxD_scaffold | http://www.ncbi.nlm.nih.gov/nuccore/KU866087 | Publicly available at NCBI GenBank (accession no: KU866087) |

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
