## [Decision Letter]

Thank you for submitting your article "Reorganisation of Hoxd regulatory landscapes during the evolution of a snake-like body plan" for consideration by *eLife*. Your article has been reviewed by three peer reviewers, and the evaluation has been overseen by a Reviewing Editor (Robb Krumlauf) and Janet Rossant as the Senior Editor. The following individuals involved in review of your submission have agreed to reveal their identity: Hugo Parker (Reviewer #1); Jacqueline Deschamps (Reviewer #2); Axel Meyer (Reviewer #3).

The reviewers have discussed the reviews with one another and the Reviewing Editor has drafted this decision to help you prepare a revised submission.

Summary:

A large amount of work is brought together here through analysis of the *Hoxd* cluster in an effort to explain why a snake ends up looking like a snake and not like a mouse. Although *Hox* gene cluster architecture is extremely conserved among different classes of vertebrates, their body plans are surprisingly diverse. This "disconnect" between genomic conservation and phenotypic diversity is intriguing and still largely not understood. Snakes are secondarily simplified vertebrates that possess a derived/unusual tetrapod body plan and alterations in *Hox* gene function and regulation have been suggested to play an important role in bringing about that change. This study relies extensively on the known principles of *Hox* regulation as previously characterized by the author's group in mouse. Extrapolating from this mouse "ground state" the study presents a descriptive molecular genetic comparative analyses of regulatory mechanisms between a model (mouse) and non-model organism (snake), focusing on mesodermic enhancer activity. This work provides interesting advances towards answering that the central question through a detailed analysis of the regulation of the snake *Hoxd* cluster. It contains large and detailed datasets, including BAC transgenesis, ChIP-seq and 4C-seq.

The manuscript is, in principle of sufficient quality and broad interest to warrant publication in *eLife*, but several central questions and concerns need to be addressed which may require additional data. These issues fall into three main areas: 1) the nature of the mesoderm; 2) the postulated location of mesodermal enhancers; and 3) analysis of the exaptation finding.

More specifically:

1).The authors should make it more explicit that the differently regulated mesodermal *hoxd4* expression domains are the lateral plate and intermediate mesoderm, and that these be termed 'ventral mesoderm' in the manuscript. It is more appropriate to call the mesoderm where *HoxD* genes are regulated differently in snake and other vertebrates "ventral mesoderm" (meaning lateral plate and intermediate mesoderm together, in fact the tissues where limbs and external genitals arise from) instead of other versions of this mesoderm used through the manuscript ("upper mesodermal derivatives", " trunk mesoderm"). From the expression patterns documented in Figure 2 and Figure 4 and Figure 4—figure supplement 1, it is quite clear from the whole mounts that the difference in regulatory potential from inside the cluster between snake and other vertebrates concerns ventral mesoderm. This is clear even in the absence of identifying the mesodermal components involved upon sectioning the embryos. Dealing with the presence of a ventral mesoderm enhancer in the telomeric desert flanking mouse *HoxD* then is not at odds with the data described in Spitz et al., 2001. In that paper expression of a "scanning lacZ reporter" replacing all *HoxD* genes remains expressed in the ventral most mesoderm. This addresses any concern about the specific domains of mesodermal expression in the transgenics and mutants. It might also be useful if the authors would label this in Figure 2 for those unfamiliar with mouse embryos.

2) All comments regarding whether a ventral mesodermal enhancer is likely to exist in the snake telomeric desert (a situation that cannot be tested in snakes), and whether it is legitimate to say that snake mesoderm-specific enhancers are mostly located within the *HoxD* cluster itself rather than outside raise the point of why don't the authors invoke their most parsimonious hypothetical scenario: telomeric mesodermal enhancers might exist both in snake (without being identified so far) and in other vertebrates, accounting for *HoxD* expression in ventral mesoderm in these species, a potential conserved regulatory event. In tetrapods this mesodermal expression of *HoxD* would be repressed at the initiation of the second wave of *HoxD* expression to ensure distal limb morphogenesis. In snakes, which are limbless, this negative control would not take place (discussion and ventral mesoderm expression of *HoxD* remains. There would thus not be snake-specific mesoderm enhancers inside the snake *HoxD* cluster, but a loss of repression. The authors need to soften their assertion about the absence of a ventral mesodermal enhancer/s in the snake telomeric desert, along these lines.

3) The exaptation finding is potentially most illuminating from an evolutionary perspective. However, activity of snake enhancers in mouse is not necessarily a reliable readout of their activity in snake, due to regulatory systems drift. However, as testing these elements in a reporter assay in a lizard or snake is too much to ask, perhaps tissue specific histone marks such as H3K27ac could be informative. This approach was recently used by Infante et al. 2015 (Developmental Cell 35.107-119) for Anolis limb and genital tissue. The authors could look through the ChIP-seq data from this Anolis study, available on-line, to see whether Anolis homologs of their elements show up. At least one of the homologous Anolis enhancers, as well as a homolog from another snake, could also be tested in mouse, as was done by Infante et al. for the Tbx4 HLEB enhancer. This would greatly enhance the evolutionary conclusions about the changing activities of these enhancers.

---

## [Author Response]

1) The authors should make it more explicit that the differently regulated mesodermal hoxd4 expression domains are the lateral plate and intermediate mesoderm, and that these be termed 'ventral mesoderm' in the manuscript. It is more appropriate to call the mesoderm where HoxD genes are regulated differently in snake and other vertebrates "ventral mesoderm" (meaning lateral plate and intermediate mesoderm together, in fact the tissues where limbs and external genitals arise from) instead of other versions of this mesoderm used through the manuscript ("upper mesodermal derivatives", " trunk mesoderm"). From the expression patterns documented in Figure 2 and Figure 4 and Figure 4—figure supplement 1, it is quite clear from the whole mounts that the difference in regulatory potential from inside the cluster between snake and other vertebrates concerns ventral mesoderm. This is clear even in the absence of identifying the mesodermal components involved upon sectioning the embryos. Dealing with the presence of a ventral mesoderm enhancer in the telomeric desert flanking mouse HoxD then is not at odds with the data described in Spitz et al., 2001. In that paper expression of a "scanning lacZ reporter" replacing all HoxD genes remains expressed in the ventral most mesoderm. This addresses any concern about the specific domains of mesodermal expression in the transgenics and mutants. It might also be useful if the authors would label this in Figure 2 for those unfamiliar with mouse embryos.

We agree with the reviewers that a more detailed description of which part of the trunk mesoderm expression is affected in the different lines was necessary. To this aim, we have added a Supplementary Figure (Figure 2—figure supplement 1) where we show a transversal section of both the Del(*Attp-Sb3*), The human BAC and the snake BAC transgenic embryos, as well as of a control embryo. It appears that the isolated human cluster does not have the ability to drive expression in any of the trunk mesoderm at this developmental stage and *Hoxd4* expression is clearly restrictedto the neural tube. The deletion of the mouse telomeric desert also lacks normal ventral mesoderm expression. To our understanding, the ventral-most somitic mesoderm expression is also affected in addition to the more clearly absent intermediate and lateral mesoderm. The text has been changed accordingly (subsection “Reshuffling mesodermal enhancers”) in order to take into account this new figure and the term “ventral mesoderm” is then used throughout the manuscript to describe ventral somitic mesoderm, as well as intermediate and lateral mesoderm.

We have also highlighted this ventral mesoderm region in Figure 2.

2) All comments regarding whether a ventral mesodermal enhancer is likely to exist in the snake telomeric desert (a situation that cannot be tested in snakes), and whether it is legitimate to say that snake mesoderm-specific enhancers are mostly located within the HoxD cluster itself rather than outside raise the point of why don't the authors invoke their most parsimonious hypothetical scenario: telomeric mesodermal enhancers might exist both in snake (without being identified so far) and in other vertebrates, accounting for HoxD expression in ventral mesoderm in these species, a potential conserved regulatory event. In tetrapods this mesodermal expression of HoxD would be repressed at the initiation of the second wave of HoxD expression to ensure distal limb morphogenesis. In snakes, which are limbless, this negative control would not take place (discussion and ventral mesoderm expression of HoxD remains. There would thus not be snake-specific mesoderm enhancers inside the snake HoxD cluster, but a loss of repression. The authors need to soften their assertion about the absence of a ventral mesodermal enhancer/s in the snake telomeric desert, along these lines.

We initially concluded that most mesodermal enhancers were mostly located within the gene cluster following the results obtained with the BAC transgenic experiment. We agree that even though we can conclude that sequences within the snake cluster are sufficient to drive expression in the ventral mesoderm of the mouse, this experiment does not exclude that other enhancers are located outside of the cluster. However, the H3K27 acetylation ChIP in mouse and snake trunk shows a considerable decrease of peaks in the regulatory gene deserts of the snake. Therefore, although the complete absence of enhancers in the telomeric gene desert cannot be claimed without further experiments, it appears that, at the very least, a significant reduction of mesodermal enhancers outside of the cluster may have occurred. We have changed the text to correct the premature conclusion on the location of the snake mesodermal enhancers (subsection “Bimodal regulation in the snake HoxD regulatory locus”).

In our view, the conclusion that most trunk mesodermal enhancers are present within the snake cluster, unlike the mouse, remains the most parsimonious. In the mouse, mesodermal enhancers have been described to be present within the cluster and snakes seem to have found an evolutionary advantage in increasing the regulatory potential of sequences in close proximity to their target genes. Such a scenario would account for the use of a preexisting regulatory strategy that would simply be reinforced in the case of the snake. In contrast, mouse limb enhancers have only been described outside of the cluster and therefore it is unlikely that a limbless organism such as the snake would select for the re-allocation of limb enhancers within the cluster. It is therefore possible that snake *Hoxd* genes have suffered distinct regulatory changes perhaps depending on whether such changes were selected after positive or relaxed selection. We have changed the text to more extensively discuss this hypothesis and have toned down our conclusions.

*3) The exaptation finding is potentially most illuminating from an evolutionary perspective. However, activity of snake enhancers in mouse is not necessarily a reliable readout of their activity in snake, due to regulatory systems drift. However, as testing these elements in a reporter assay in a lizard or snake is too much to ask, perhaps tissue specific histone marks such as H3K27ac could be informative. This approach was recently used by Infante et al. 2015 (Developmental Cell 35.107-119) for Anolis limb and genital tissue. The authors could look through the ChIP-seq data from this Anolis study, available on-line, to see whether Anolis homologs of their elements show up. At least one of the homologous Anolis enhancers, as well as a homolog from another snake, could also be tested in mouse, as was done by Infante et al. for the Tbx4 HLEB enhancer. This would greatly enhance the evolutionary conclusions about the changing activities of these enhancers.*

We have followed this interesting suggestion and have analyzed the recently published acetylation tracks of limb and genitalia from lizards and mice (Infante et al., 2015). Interestingly, we find that, like in the mouse, the lizard Island I is enriched for H3K27ac in limb samples (see Figure 6). Some level of acetylation is also present in lizard genitals. This prompted us to assay the lizard Island I sequence in an enhancer assays, together with the chicken counterpart (Figure 5, top). We find that, while the chicken Island I has a limb-only function, thus looking very much like the mouse sequence, the lizard counterpart contains both limb and genital regulatory specificities, even though the genital expression was much weaker than in the limbs. Since the chicken is more closely related to squamates (lizards and snakes) than to mammals, these results suggest that an ancestral Island I enhancer had a limb-only specificity, which was subsequently co-opted to a genital specificity in squamates. Of course all these assays were performed in a mouse background, yet the correspondence between reporter activity experiments in the mouse and the previously published H3K27ac ChIP-seq in other species re-enforces the relevance of this experimental approach.

These new results are described in the revised text.

Author response image 1.Figure 1 H3K27ac profiles in limb and genitals over Island I of mouse (up) and lizard (bottom).The Island I is shown as a black box under the profiles and sequence conservation as assessed by UCSC Blat is shown underneath for different vertebrate species. The mouse Island I is enriched only in limbs. The lizard Island I although clearly acetylated in limbs also displays some level of acetylation in genitals.**DOI:**
http://dx.doi.org/10.7554/eLife.16087.018